# Closed-loop stimulation of temporal cortex rescues functional networks and improves memory

Youssef Ezzyat *et al.*[#]

Memory failures are frustrating and often the result of ineffective encoding. One approach to improving memory outcomes is through direct modulation of brain activity with electrical stimulation. Previous efforts, however, have reported inconsistent effects when using open-loop stimulation and often target the hippocampus and medial temporal lobes. Here we use a closed-loop system to monitor and decode neural activity from direct brain recordings in humans. We apply targeted stimulation to lateral temporal cortex and report that this stimulation rescues periods of poor memory encoding. This system also improves later recall, revealing that the lateral temporal cortex is a reliable target for memory enhancement. Taken together, our results suggest that such systems may provide a therapeutic approach for treating memory dysfunction.

. Correspondence and requests for materials should be addressed to M.J.K. (email: kahana@sas.upenn.edu). [#]A full list of authors and their affliations appears at the end of the paper.

Research on human episodic memory has shown that whether information is remembered or forgotten depends on neural events that transpire during encoding. Spectral power recorded using intracranial electrophysiology[1] and blood-oxygen-level-dependent functional magnetic resonance imaging (fMRI) signal[2] show that activity in many cortical and subcortical regions differentiates learned information that is likely to be remembered from information that is likely to be forgotten. Differences in neural activity during encoding therefore predict intra-individual variability in later memory performance, suggesting that modulating neural activity when the brain is unlikely to encode successfully could improve overall performance by rescuing network activity.

A promising tool for modulation of neural activity is direct brain stimulation, in which electrical current is applied via electrodes implanted on or directly in the brain parenchyma. Direct brain stimulation is a standard tool in the treatment of motor dysfunction in Parkinson's disease and seizure onset in epilepsy[3–6], and has recently been explored as a therapy for psychiatric conditions[7,8]. Direct brain stimulation treatment commonly involves continuous (i.e., open-loop) high-frequency stimulation, although recent work has suggested improved effectiveness when applying stimulation in response to specific brain states (i.e., closed-loop[9,10]).

Several studies have used direct brain stimulation of the human hippocampus and medial temporal lobes (MTLs), core regions of the brain's memory network, to modulate neural activity and performance during memory tasks. These studies used open-loop designs, meaning that stimulation was not delivered in response to ongoing neural activity. Although some studies using this approach have suggested that such stimulation improves memory[11–14], others have failed to show improvements or have shown disruption[15–20]. This literature shows that direct open-loop stimulation of the hippocampus and MTL is unlikely to reliably improve memory, and suggests that stimulating other regions in the memory network may be more effective for memory enhancement.

Encoding activity in the left lateral temporal cortex (including the middle portions of the inferior, middle, and superior temporal gyri) measured with fMRI[2] and intracranial electro-encephalography (iEEG)[1,21,22] predicts memory performance, and stimulation mapping of this area has been shown to evoke memory-like recollective phenomena[23,24]. Prior invasive[25–27] and non-invasive[28–30] stimulation studies also suggest this region to be a prime target for memory modulation. However, these studies did not characterize the effect of lateral temporal cortex stimulation in comparison with other targets. The previous work also did not use stimulation to rescue intervals of poor memory encoding by timing stimulation based on recordings of ongoing neural activity.

Here we evaluate the use of lateral temporal cortex as a stimulation target by deploying a closed-loop architecture for sensing and stimulating the brain during a memory task. We compare the effect of lateral temporal cortex stimulation to both a within-subject non-stimulated condition and an independent non-lateral temporal stimulation group. We use multivariate classifiers that are individualized to each subject to decode neural activity during memory encoding and trigger stimulation online in response to patterns of neural activity that are associated with later forgetting. By using classification within a closed-loop system, we maximize sensitivity to detect poor encoding states and account for the fact that direct brain stimulation has distributed effects on physiology[31,32] that depend on the state of the brain at the time of stimulation delivery[33–37].

## Results

**Classifying recall probability from neural activity.** We recruited 25 neurosurgical patients undergoing clinical monitoring for epilepsy to participate in sessions of a delayed free recall memory task (Fig. 1a). Subjects performed at least three record-only sessions of free recall from which we trained a multivariate classifier to discriminate patterns of neural activity during encoding that predicted memory. We fit penalized logistic regression classifiers to record-only data from each subject, producing a set of model weights that map features of iEEG activity to an output probability of later word recall (Fig. 1b).

We then used this model in subsequent sessions to decode the probability of recall from neural activity online during the

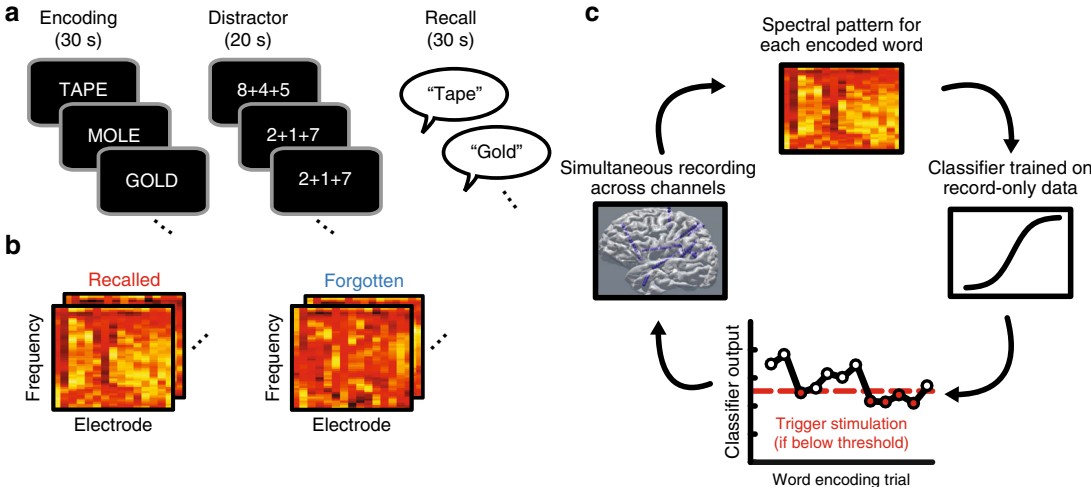

**Fig. 1** Closed-loop approach. **a** For each list of the free recall task, subjects encoded 12 nouns presented sequentially, followed by an arithmetic distractor and the verbal recall phase. Subjects performed at least three sessions of record-only free recall. **b** After the record-only sessions, we use spectral decomposition to measure power at a set of frequencies ranging from 3 to 180 Hz for each encoded word. We used the patterns of spectral power across electrodes to train a penalized logistic regression classifier to discriminate encoding activity during subsequently recalled words from subsequently forgotten words. **c** In later closed-loop sessions, we applied spectral decomposition to each word encoding period while subjects performed the task. This produced a set of frequency × electrode features to which we applied the classifier trained on the record-only data. If the resulting estimated probability of recall was below 0.5, we triggered 500 ms of stimulation to either the lateral temporal cortex or a control target

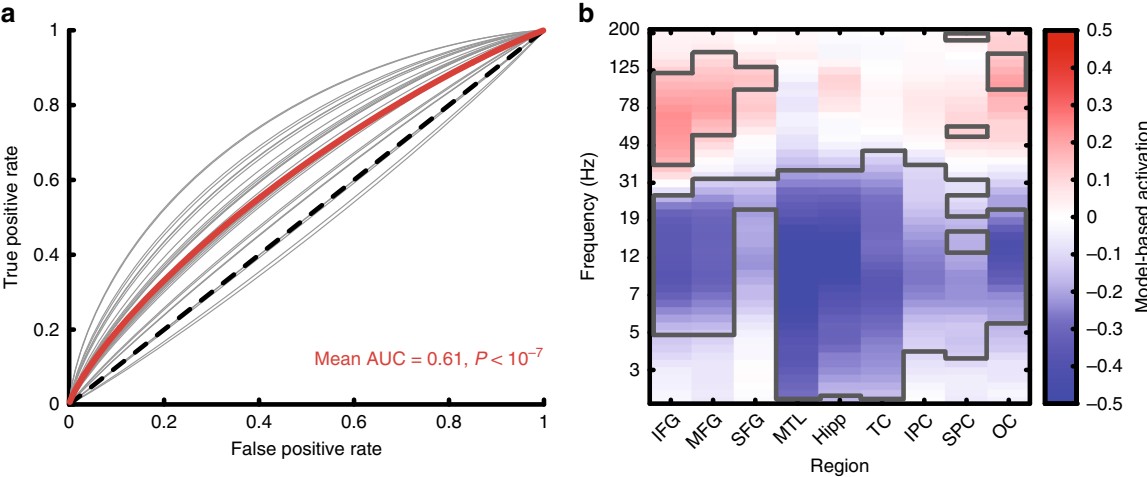

**Fig. 2** Classifier performance and features. **a** Receiver operating characteristic (ROC) curves showing performance of a record-only classifier tested on NoStim lists during the closed-loop sessions (gray lines). Each curve displays the ROC for a unique subject–stimulation site. The group average is shown in red (mean AUC = 0.61, $P < 10^{-7}$ by a one-sample $t$-test, $N = 29$ unique stimulation targets). **b** Forward model-based estimates of feature importance. For each electrode region by frequency feature, we computed a forward model-based estimate of the feature's contribution to the classification decision[34]. This analysis shows that the classifier used increased high-frequency power combined with decreased low-frequency power to predict successfully recalled words (clusters significant at the $P < 0.05$ level by a one-sample $t$-test are outlined in gray, $N = 25$ record-only classifiers)

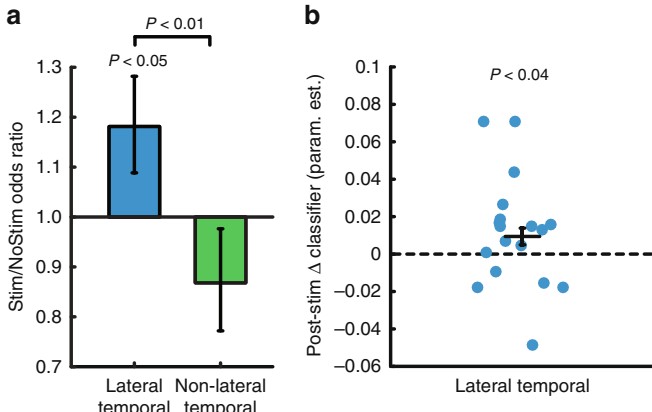

**Fig. 3** Stimulation affects behavior and physiology. **a** Stimulation delivered to lateral temporal cortex targets increased the probability of recall compared to matched unstimulated words in the same subject ($P < 0.05$) and stimulation delivered to Non-lateral temporal targets in an independent group ($P < 0.01$). **b** The change in classifier output post-lateral temporal cortex stimulation was greater than for matched intervals on NoStim lists. Data are presented as model parameter estimate ± SE. Lateral temporal cortex $N = 18$; Non-lateral temporal $N = 11$

encoding phase of the task (Fig. 1c). On Stim lists, if the predicted probability of recall was below 0.5, the system triggered 500 ms of bipolar stimulation across an adjacent pair of channels. We predicted that this would disrupt poor encoding states and improve memory performance. On NoStim lists, we used the classifier to estimate recall probabilities but did not trigger stimulation. This allowed us to control for the memory state when assessing the effect of stimulation on recall probability of stimulated items. It also allowed us to determine whether the record-only classifier generalized to the new closed-loop session. We found that these classifiers reliably discriminated recalled from not recalled words in the new session (NoStim list area under the curve (AUC) = 0.61, $t(28) = 7.5$, $P < 10^{-7}$ by a one sample $t$-test, Fig. 2a). To determine what information the model used to classify encoding trials we computed a Forward Model for each patient using the classifier weights and neural encoding

activity[34,38]. Across subjects, this showed that the classifiers relied on increased high-frequency activity (HFA) across the brain (frontal, temporal, parietal, and occipital cortex), along with widespread decreases in low-frequency activity to predict words likely to be recalled (Fig. 2b; clusters outlined in gray were significant at the $P < 0.05$ level by a one-sample $t$-test).

**Stimulation increased the probability of word recall**. We used a generalized linear mixed effects (GLME) model (logistic regression with a binomial distribution) to model the individual trial-level recalled/not recalled data across subjects, to determine how stimulation affected the probability of word recall. The linear mixed effects approach allows us to estimate the effect of stimulation on probability of recall while accounting for heterogeneity in the amount of data collected from each subject. The resulting parameter estimates reflect the change in odds associated with each condition (Stim/NoStim and Lateral temporal cortex/Non-lateral temporal). Stimulation of lateral temporal cortex increased the odds of recalling stimulated words compared with matched non-stimulated words (odds ratio = 1.18, $t(3,828) = 2.04$, $P = 0.04$, Fig. 3a) and compared with Non-lateral temporal stimulation ($t(5,864) = 2.60$, $P = 0.009$). Stimulation led to a nonsignificant decrease in recall odds in the Non-lateral temporal group (odds ratio = 0.87, $t(2,036) = -1.20$, $P = 0.23$). The stimulation targets that led to increased memory performance in the lateral temporal cortex group clustered in the middle portion of the left middle temporal gyrus (Fig. 4). Using a log-binomial model[39] to estimate the relative change in recall probability, we found that lateral temporal cortex stimulation increased the relative probability of item recall by 15% ($t(3828) = 2.31$, $P = 0.02$, Supplementary Table 1). Subjects stimulated in lateral temporal cortex were also more likely to remember unstimulated items that flanked the stimulation ($t(960) = 2.46$, $P = 0.01$; interaction with Non-lateral temporal stimulation $t(1,406) = 2.48$, $P = 0.01$).

We observed these different stimulation-related memory outcomes in spite of the fact that the groups were matched in several ways. The proportion of words on which the closed-loop classifier triggered stimulation was roughly 0.50 (as intended) and did not differ between the lateral temporal cortex and Non-lateral temporal groups (lateral temporal cortex = 0.52, Non-lateral

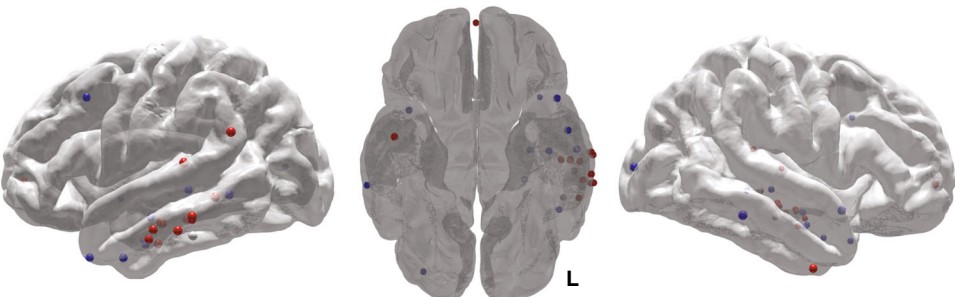

**Fig. 4** Stimulation targets rendered on an average brain surface. Targets showing numerical increase/decrease in free recall performance are shown in red/blue. Memory-enhancing sites clustered in the middle portion of the left middle temporal gyrus (coordinate range $X : -67$ to $-47$; $Y : -51$ to $-1$; $Z : -33$ to 8)

temporal = 0.50, $t(5,698) = 0.30$, $P = 0.71$ by linear mixed effects model). Stimulation also did not differ between the groups as a function of word position within the list (position × group interaction $t(5,696) = -0.84$, $P = 0.40$). The groups showed similar memory performance during the record-only sessions (lateral temporal cortex = 26.1%, Non-lateral temporal = 29.8%, $t(24) = -0.87$, $P = 0.39$ by two-sample $t$-test). We also analyzed subsequent memory effects (SMEs) at the stimulated electrodes in the record-only data, to determine whether the stimulated electrodes showed different contributions to memory performance across the groups. We found that power across the frequency spectrum (2–200 Hz) was similarly predictive of memory for the lateral temporal cortex and Non-lateral temporal groups (Supplementary Figure 1a, b), indicating no difference between groups in the stimulated electrode's involvement in encoding processes. We also did not find any relation between the size of the SME at the stimulated electrode and stimulation's effect on behavior (Supplementary Figure 1c).

**Neural evidence for improved encoding following stimulation**. We next asked how lateral temporal cortex stimulation influenced neural activity to support memory encoding by comparing HFA (70–200 Hz) between Stim and NoStim conditions. High-frequency power predicts memory success[1,37,40,41] and is thought to reflect excitation of neural populations engaged during cognition[42,43]. When comparing evoked HFA following stimulation to matched unstimulated periods in electrodes placed in the same lobe as the stimulation target, we did not observe significant increases in HFA power following lateral temporal cortex stimulation (Supplementary Fig. 2). This suggests that lateral temporal cortex stimulation did not reliably modulate activity in these target regions when aggregating data across subjects, which was not surprising given that stimulation has heterogeneous effects on downstream targets[31,32], and that our subjects each possessed a unique recording montage. We reasoned that a more sensitive approach would be to assess the effect of stimulation on neural activity using a model that accounts for this between-subject variability.

The classifier that we trained on each subject's record-only data is an individualized model relating neural activity in that person to a probability of recall, so we next asked whether these classifiers could be used to decode the effects of stimulation on neural activity. We assessed this by computing the change in whole-brain classifier output for consecutive words ($w_{i+1}$ and $w_i$, for all stimulated words $i$). We did the same for matched words on NoStim lists and compared these Δ classifier values between the Stim and NoStim conditions for the lateral temporal cortex group using a mixed model. This analysis showed increases in classifier output following stimulation compared to the control non-stimulated condition (estimate = 0.01, $t(1,404) = 2.06$,

$P = 0.04$ by linear mixed effects model, Fig. 3b), suggesting that stimulation's influence on neural activity was evident when using the whole-brain classifier to decode the change in neural activity post-stimulation.

## Discussion

In this study we used closed-loop direct brain stimulation of the human lateral temporal cortex to improve episodic memory performance. By demonstrating that lateral temporal cortex stimulation enhances memory, our findings show that stimulating neural populations outside of the MTL can reliably improve memory outcomes. We used a novel method for applying stimulation by developing patient-specific models of neural activity that were used to classify memory-related brain states. Stimulation was then triggered to intervene in response to intervals when memory function was predicted to be poor. The data suggest a strategy for memory modulation that has applications for the treatment of memory dysfunction.

In observing that closed-loop stimulation improves performance in the free recall task, we show that stimulation can reliably benefit memory if it is timed to rescue periods of poor memory function. In order to predict when subjects would be in poor memory encoding states, we used data collected during performance of the free recall task to create models relating whole-brain neural activity to memory outcomes. We found that these models predicted variability in memory performance when used in subsequent closed-loop sessions, suggesting that our approach identified patterns of neural activity that predicted memory performance in a stable manner over time. Stimulating in this way led to both increased model estimates of recall probability and increased memory performance, suggesting that the system improved memory by rescuing periods of poor memory function.

We also identify the lateral temporal cortex as a prime target for modulating memory encoding, moving beyond prior work that has attempted to influence episodic memory by stimulating the hippocampus and MTL [12,14,15,20]. Several previous studies have investigated the role of the lateral temporal lobes in episodic memory using direct brain stimulation[23,25–27]; our work further demonstrates that closed-loop stimulation of this region can reliably alter memory-related neural activity and memory performance. As our classifier integrated activity over electrodes implanted across the brain, the findings suggest that lateral temporal cortex stimulation improves encoding by influencing a distributed memory-supporting network. This is consistent with the idea that stimulation of cortical areas at or near the surface of the brain can affect neural activity in downstream-connected targets and suggests applications of our approach to non-invasive methods[44].

Functional and structural connectivity between the stimulation target and the rest of the brain may be useful in predicting the effect of lateral temporal cortex stimulation on memory, as well as the brain areas that are likely to be modulated by stimulation. Connectivity defined using resting-state fMRI, for example, has been shown to predict the effects of invasive and non-invasive stimulation by identifying stimulation targets based on their membership in large-scale functional networks[45]. Brain-wide networks that underlie successful memory encoding show theta-band iEEG synchrony, which could also be used to identify stimulation targets in the memory network[46]. Measures of network controllability based on connectivity could also be used to predict the extent to which a stimulation target can control functional dynamics in the memory network[47].

Our system also provides a framework for developing therapies to treat memory dysfunction. There is evidence that disorders such as Alzheimer's disease demonstrate network abnormalities resulting from inhibitory dysfunction[48] that can manifest early in disease progression. Such changes in the balance between inhibitory and excitatory activity within a neural population can lead to alterations in the $1/f$ slope recorded in the local field potential[49], with increased inhibition resulting in a steeper slope. We found that the classifiers in our study relied on simultaneously increased HFA and decreased low-frequency activity (i.e., a decrease in the $1/f$ slope) to differentiate good and poor memory states. Given that stimulation increased classifier estimates of memory states, the data suggest modulation of the excitatory/ inhibitory balance to be a potential mechanism of action. Future work could evaluate directly whether and how closed-loop stimulation changes the excitatory/inhibitory balance in local and remote sites in the memory network.

The current findings demonstrate that intervals of poor memory encoding can be identified online and rescued with targeted stimulation to the lateral temporal cortex. Using this closed-loop stimulation approach in patients with epilepsy, we provide proof of concept for the therapeutic treatment of memory dysfunction.

## Methods

**Participants.** Twenty-five patients undergoing intracranial electroencephalographic monitoring as part of clinical treatment for drug-resistant epilepsy were recruited to participate in this study. Data were collected as part of a multi-center project designed to assess the effects of electrical stimulation on memory-related brain function. Data were collected at the following centers: Thomas Jefferson University Hospital (Philadelphia, PA), University of Texas Southwestern Medical Center (Dallas, TX), Emory University Hospital (Atlanta, GA), Dartmouth-Hitchcock Medical Center (Lebanon, NH), Hospital of the University of Pennsylvania (Philadelphia, PA), and Mayo Clinic (Rochester, MN). The research protocol was approved by the Institutional Review Board at each hospital and informed consent was obtained from each participant. Electrophysiological data were collected from electrodes implanted subdurally on the cortical surface as well as depth electrodes within the brain parenchyma. In each case, the clinical team determined the placement of the electrodes so as to best localize epileptogenic regions. Subdural contacts were arranged in both strip and grid configurations. The types of electrodes used for recording and stimulation varied across the data collection sites in accordance with the preferences of the clinicians at each institution. Across the sites, the following models of depth and surface (strip/grid) electrodes were used (electrode diameters in parentheses): PMT Depthalon (0.8 mm); AdTech Spencer RD (0.86 mm); AdTech Spencer SD (1.12 mm); AdTech Behnke-Fried (1.28 mm); AdTech subdural strips and grids (2.3 mm).

**Verbal memory task.** Each subject participated in a delayed free-recall task in which they were instructed to study lists of words for a later memory test; no encoding task was used. Lists were composed of 12 words chosen at random and without replacement from a pool of high frequency English nouns (http://memory. psych.upenn.edu/WordPools). Each word remained on the screen for 1,600 ms, followed by a randomly jittered 750–1,000 ms blank inter-stimulus interval.

Immediately following the final word in each list, participants performed a distractor task (to attenuate the recency effect in memory, length = 20 s) consisting of a series of arithmetic problems of the form $A + B + C = ??$, where A, B, and C were randomly chosen integers ranging from 1 to 9. Following the distractor task

participants were given 30 s to verbally recall as many words as possible from the list in any order; vocal responses were digitally recorded and later manually scored for analysis. Each session consisted of 25 lists of this encoding-distractor-recall procedure. Some subjects completed sessions of the free recall task using categorized word lists, which were included in the electrophysiological analyses. The categorized recall task is identical to the free recall task, with the exception that the word pool was drawn from 25 semantic categories (e.g., fruit, furniture, office supplies). Each list of 12 items in the categorized version of the task consisted of four words drawn from each of three categories. Subject counts by task: $N = 17$ free recall only; $N = 2$ categorized free recall only; $N = 6$ both free and categorized recall (in separate sessions).

**Stimulation methods.** At the start of each session, we determined the safe amplitude for stimulation using a mapping procedure in which stimulation was applied at 0.5 mA, while a neurologist monitored for afterdischarges. This procedure was repeated, incrementing the amplitude in steps of 0.5 mA, up to a maximum of 1.5 mA for depth contacts and 3.5 mA for cortical surface contacts. These maximum amplitudes were chosen to be below the afterdischarge threshold and below accepted safety limits for charge density[50]. For each stimulation session, we passed electrical current through a single pair of adjacent electrode contacts. As the electrode locations were determined strictly by the monitoring needs of the clinicians, we used a combination of anatomical and functional information to select stimulation sites. If available, we prioritized electrodes in lateral temporal cortex, in particular the middle portion of the middle temporal gyrus. To choose among these regions in cases in which more than one was available, we selected the electrode pair demonstrating the largest SME, in the high frequency range (70–200 Hz). In cases in which no lateral temporal cortex contacts were available, we selected an electrode pair at or near the largest SME elsewhere in the brain, targeting the hippocampus, MTL cortex, prefrontal cortex, and parietal cortex if available. Stimulation was delivered using charge-balanced biphasic rectangular pulses (pulse width = 300 μs) at (10, 25, 50, 100, or 200) Hz frequency and (0.25 to 2.00) mA amplitude (0.25 mA steps). The particular amplitude and frequency were chosen based on a pre-test in which we stimulated the brain at each parameter combination, while the patient was at rest (no experimental task). The frequency × amplitude combination that maximized the change in classifier output was used in the closed-loop memory task. In the memory task stimulation was always applied for 500 ms in response to classifier-detected poor memory states (see below). Participants performed one practice list followed by 25 task lists: lists 1–3 (plus the practice list) were used to collect baseline spectral power data for normalizing input features for the classifier; lists 4–25 consisted of 11 lists each of Stim and NoStim conditions, randomly interleaved. On NoStim lists, stimulation was not triggered in response to classifier output.

**Anatomical localization.** Cortical surface regions were delineated on pre-implant whole brain volumetric T1-weighted MRI scans using Freesurfer[51] according to the Desikan–Killiany atlas. Whole brain and high resolution MTL volumetric segmentation was also performed using the T1-weighted scan and a dedicated hippocampal coronal T2-weighted scan with Advanced Normalization Tools (ANTS)[52] and Automatic Segmentation of Hippocampal Subfields multi-atlas segmentation methods[53]. Coordinates of the radiodense electrode contacts were derived from a post-implant computed tomography and then registered with the MRI scans using ANTS. Subdural electrode coordinates were further mapped to the cortical surfaces using an energy minimization algorithm[54]. Two neuroradiologists reviewed cross-sectional images and surface renderings to confirm the output of the automated localization pipeline. Targets that were localized to the left inferior, middle, and superior temporal gyri were classified as lateral temporal cortex. Any target outside these regions was classified as Non-lateral temporal.

**Electrophysiological data processing.** Intracranial data were recorded using one of the following clinical EEG systems (depending on the site of data collection): Nihon Kohden EEG-1200, Natus XLTek EMU 128 or Grass Aura-LTM64. Depending on the amplifier and the preference of the clinical team, the signals were sampled at either 500, 1,000, or 1,600 Hz and were referenced to a common contact placed intracranially, on the scalp, or mastoid process. Intracranial electrophysiological data were filtered to attenuate line noise (5 Hz band-stop fourth order Butterworth, centered at 60 Hz). To eliminate potentially confounding large-scale artifacts and noise on the reference channel, we re-referenced the data using a bipolar montage [1]. To do so, we identified all pairs of immediately adjacent contacts on every depth, strip and grid and took the difference between the signals recorded in each pair. The resulting bipolar timeseries was treated as a virtual electrode and used in all subsequent analysis. We performed spectral decomposition (8 frequencies from 3 to 180 Hz, logarithmically spaced; Morlet wavelets; wave number = 5) for the 0–1,366 ms epoch relative to word onset. Mirrored buffers (length = 1,365 ms) were included before and after the interval of interest and then discarded to avoid convolution edge effects. The resulting time-frequency data were then log-transformed, averaged over time, and z-scored within session and frequency band across word presentation events.

**Multivariate classification**. We included only data collected in record-only sessions as input to a logistic regression classifier trained to discriminate encoding-related activity predictive of whether a word was later remembered or forgotten. We used spectral power averaged across the time dimension for each word encoding epoch (0–1,366 ms relative to word onset) as the input data (informal investigations suggested that spectral power outperformed features derived from phase-based connectivity measures). Thus, the features for each individual word encoding observation were the average power across time, at each of the eight analyzed frequencies × N electrodes. We used L2-penalization[55] and set the penalty parameter ($C$) to $2.4 \times 10^{-4}$, based on the optimal penalty parameter calculated across our large pre-existing dataset of free recall subjects[37]. We then computed AUC to quantify classifier performance and repeated this procedure for all subjects. AUC measures a classifier's ability to identify true positives while minimizing false positives, where chance AUC = 0.50. To ensure the classifier learned equally from both classes (given the imbalance between recalled and not recalled exemplars), we also weighted the penalty parameter in inverse proportion to the number of exemplars of each class[55]. We assessed the significance of each classifier within-subject using a permutation test in which we randomized the labels of recalled/not recalled events in the training data, computed AUC and repeated the randomization 1,000 times to generate a null distribution of AUCs. Classification analyses were programmed using the Matlab implementation of the LIBLINEAR library[56].

We generated a forward model for each subject[34] to assess the importance of different features to classification

$$A = \frac{\Sigma_{\mathbf{x}}\mathbf{W}}{\sigma_{\hat{\mathbf{y}}}^2}$$

where $\Sigma_{\mathbf{x}}$ is the data covariance matrix, $\mathbf{W}$ is the vector of weights obtained from the fitted classifier, $\sigma_{\hat{\mathbf{y}}}^2$ is the variance of the logit-transform of the vector of classifier outputs across all events, $\hat{\mathbf{y}}$. The magnitude and direction of the values in $\mathbf{A}$ reflect the strength of the relation between classifier output and each input feature in $\mathbf{x}$. We computed $\mathbf{A}$ separately for each subject and plot the average across subjects in Fig. 2b.

**Closed-loop decoding**. The closed-loop free recall sessions consisted of one practice list followed by 25 task lists: lists 1–3 were unstimulated and used (in addition to the practice list) to collect baseline spectral power data for normalizing the input features to the classifier; lists 4–25 consisted of 11 lists each of Stim and NoStim conditions, randomly interleaved. On Stim lists, we extracted estimates of spectral power from the 0–1,366 ms relative to each word encoding period. We assessed power using parameters identical to those used to train the record-only classifier (see Electrophysiological data processing): 8 frequencies from 3 to 180 Hz, logarithmically spaced; Morlet wavelets; wave number = 5; log-transformed; averaged over time within frequency and electrode; and z-scored based on the mean and SD of the power features collected during the baseline lists. The distribution of powers used for normalization was updated following each NoStim list. The record-only classifier was applied to the resulting frequency × electrode features to derive an estimated probability of recall. If this probability fell below 0.5, we immediately triggered 500 ms of stimulation (mean interval between stimulation onset and presentation of the next word = 962 ms ± 347 ms). On NoStim lists, spectral power features and classifier output were computed identically to Stim lists, but stimulation was disabled.

**GLME model**. We used GLME models using MatLab's fitglme.m function to estimate the effect of stimulation on memory performance. In the full model assessing the interaction of stimulation and group on memory, we modeled the recalled/not recalled status of each encoding trial for all subjects as a function of list type (Stim/NoStim) and group (Lateral temporal cortex/Non-lateral temporal), with random slopes and intercepts for the effect of stimulation for each subject and unique stimulation target site (three subjects were stimulated at two different targets in separate sessions). We included in the model stimulated words and matched words from NoStim lists. The NoStim words were matched based on whether classifier output during the closed-loop session was below threshold (i.e. stimulation would have been applied had it been a Stim list). To estimate the effect of stimulation within each of the Lateral temporal cortex/Non-lateral temporal groups we then fit the same model separately for each group without the group predictor.

We also used a GLME model to assess the influence of stimulation on memory for neighboring trials. Here we identified unrecalled stimulated words (and matched NoStim words) that were flanked in the forward and reverse directions by one unstimulated word (or words that based on classifier output would have been unstimulated, in the case of NoStim lists). We modeled the recall output using a binomial model with the following predictors: Stim/NoStim and Lateral temporal/Non-lateral temporal stimulation target, including random intercepts and slopes for the within-subject factor.

**Analysis of post-stimulation classifier**. To assess the effect of lateral temporal cortex stimulation on neural activity we used the classifier to decode the stimulation-evoked change in physiology. We fit a GLME model to predict classifier output for the word encoding period immediately following delivery of a

stimulation train. We included matched intervals from NoStim lists by identifying periods following words that would have been stimulated, and modeled the Stim/NoStim list status of the observations, with separate slopes and intercepts for each subject and unique stimulation target site.

**Statistics**. Data are presented as mean ± SEM. Unless otherwise specified, all statistical comparisons were conducted as two-tailed tests. Data distributions were either visually inspected or assumed to be normal for parametric tests. We used linear mixed effects models of the trial-level data to estimate the effect of stimulation on behavior and classifier output (e.g., Figure 3), while accounting for repeated subject and subject-stimulation location across observations. Samples sizes were chosen to meet or exceed previously reported studies of invasive lateral temporal cortex stimulation.

**Data availability**. All de-identified raw data and analysis code may be downloaded at http://memory.psych.upenn.edu/Electrophysiological_Data.

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

## Acknowledgements

We dedicate this paper in loving memory of Anastasia Lyalenko, without whose contributions this work would not have been possible (http://memory.psych.upenn.edu/Anastasia_Lyalenko_Memorial_Fund). This work was supported by the DARPA Restoring Active Memory (RAM) program (Cooperative Agreement N66001-14-2-4032). We thank Blackrock Microsystems for providing neural recording and stimulation equipment. We are indebted to the patients and their families for their participation and support. The views, opinions, and/or findings contained in this material are those of the authors and should not be interpreted as representing the official views or policies of the Department of Defense or the U.S. Government.

## Author contributions

Y.E., D.S.R., and M.J.K. designed experiments. Y.E. analyzed the data. P.A.W., D.F.L., A.K, A.A., A.B., C.S.I., M.A.G., and M.T.K. performed data collection, recording, and annotation of behavioral responses. I.P. programmed the closed-loop system. M.R.S., A.D.S., B.C.L., R.E.G., B.C.J., K.A.D., and G.A.W. recruited participants and provided general assistance. J.M.S., R.G., and S.R.D. localized the electrodes. Y.E. and M.J.K. wrote the manuscript. All authors provided feedback on the manuscript. D.S.R. and M.J.K. supervised the research.

## Additional information

**Competing interests:** B.C.J. receives research funding from NeuroPace and Medtronic not relating to this research. M.J.K. and D.S.R. are in the process of organizing Nia Therapeutics, LLC ("Nia"), a company intended to develop and commercialize brain stimulation therapies for memory restoration. Currently, Nia has no assets and has not commenced operations. M.J.K. and D.S.R. each holds a greater than 5% equity interest in Nia. All other authors declare no competing financial interests.

Youssef Ezzyat[1], Paul A. Wanda[1], Deborah F. Levy[1], Allison Kadel[1], Ada Aka[1], Isaac Pedisich[1], Michael R. Sperling[2], Ashwini D. Sharan[3], Bradley C. Lega[4], Alexis Burks[4], Robert E. Gross[5], Cory S. Inman[5], Barbara C. Jobst[6], Mark A. Gorenstein[6], Kathryn A. Davis[7], Gregory A. Worrell[8], Michal T. Kucewicz[8], Joel M. Stein[9], Richard Gorniak[10], Sandhitsu R. Das[7], Daniel S. Rizzuto[1] & Michael J. Kahana[1]

[1]Department of Psychology, University of Pennsylvania, 433 South University Avenue, Philadelphia, PA 19104, USA. [2]Deparment of Neurology, Thomas Jefferson University Hospital, 900 Walnut Street, Philadelphia, PA 19107, USA. [3]Department of Neurosurgery, Thomas Jefferson University Hospital, 900 Walnut Street, Philadelphia, PA 19107, USA. [4]Department of Neurosurgery, University of Texas, Southwestern, 5323 Harry Hines Boulevard, Dallas, TX 75390, USA. [5]Department of Neurosurgery, Emory University Hospital, 1365 Clifton Road NE, Atlanta, GA 30322, USA. [6]Department of Neurology, Dartmouth-Hitchcock Medical Center, 1 Medical Center Drive, Lebanon, NH 03756, USA. [7]Department of Neurology, Hospital of the University of Pennsylvania, 3400 Spruce Street, Philadelphia, PA 19104, USA. [8]Department of Neurology, Mayo Clinic, 200 1st Street SW, Rochester, MN 55905, USA. [9]Department of Radiology, Hospital of the University of Pennsylvania, 3400 Spruce Street, Philadelphia, PA 19104, USA. [10]Department of Radiology, Thomas Jefferson University Hospital, 900 Walnut Street, Philadelphia, PA 19107, USA

