## [Peer Review File · Nature Communications]

Reviewers' comments:

Reviewer #1 (Remarks to the Author):

The manuscript by Ezzyat et al. investigates the effect of lateral temporal cortex stimulation on memory performance in patients with implanted electrodes. The manuscript follows up on previous findings suggesting differences in neural activity during memory encoding for subsequently remembered words (termed the subsequent memory effect). The authors designed a classifier to identify these periods. Having identified these periods, they then performed stimulation during encoding of periods that the classifier identified as unlikely to be well encoded. The authors find a modest boost in performance for stimulating encoding vs. non-stimulated encoding trials. The authors conclude that stimulation outside of the hippocampus, for example, in lateral temporal cortex, can enhance memory.

Assessment: The approach here is novel, in other words, using a classifier and employing this information to stimulate locations outside of the hippocampus. I do have some concerns though with the current version of the manuscript. For one, the statistics are not adequate. I also found the regions stimulated, when not in lateral temporal cortex, poorly defined. Stimulation itself, and how it was performed, was not very clear. Finally, though convincing, the effects size is fairly modest for a large sample study like this. I detail some of these concerns below:

1. The authors should perform an ANOVA (after checking that the assumptions are met). Or, a non-parametric version of an ANOVA. Factors to include would be stimulation vs. non-stimulation, region, and whether free recall involved categorized or uncategorized words. Given the sample size, it seems that the authors could perform a logistic regression and include single trials as well as a mixed-effects ANOVA with the dependent variable being performance for each condition for each subject. Currently, though the statistics are not clear. The authors should also include information on the effects size.
2. Can the authors describe the nature of the stimulation in more detail? The details are relatively sparse but it appears to be gamma-band stimulation which varied in frequency by patient. How was the frequency selected? What were the results on the underlying electrophysiological signals from stimulation? Suppl. Figure 1 would be enhanced by comparing stimulated and non-stimulated trials. The tilt effect is not clear and thus it is not obvious why this is included.
3. The authors mention they also stimulate outside of the lateral temporal cortex. How exactly were these sites selected in terms of subsequent memory effects (largest? What frequency? Etc). Were the site selections always functionally driven? If so, how?
4. The authors argue that their stimulation effects are also whole brain. This is not clear at all from the data presented. What did recording electrodes look like outside of the stimulated area? If the authors wish to argue for whole brain effects, they should probably frame this issue in more detail, which has been discussed in past papers in some detail (e.g., Fell et al. 2013 Brain Stimulation, Kim et al. 2016 NLM).

MINOR

What was the electrode diameter?

Reviewer #2 (Remarks to the Author):

This paper describes findings from an experiment that used intracranial stimulation of patients performing free recall tasks. The goal was to determine whether the lateral temporal cortex is a viable target for stimulation to improve human memory. The experiment worked in a closed-loop manner. Several sessions of intracranial EEG (iEEG) were collected during free recall performance to train a classifier to relate brain state measured with the iEEG activity to subsequent-memory performance. This allowed the algorithm to identify when the brain was in a poor encoding state and intervene with stimulation of the lateral temporal cortex, or stimulation of control sites outside the temporal cortex.

This paper represents an excellent combination of rigorous methods, a unique patient population, and strong behavioral task to measure memory storage. This paper is a bit more exploratory than one in which competing hypotheses are distinguished. But I would expect it to have a large impact (i.e., highly cited) given the propensity for the field to focus on MTL (and the parietal and prefrontal hubs to which it connects). This paper is highly suitable for Nature Communications as it represents a new discovery with impacts in the fields of neurology, neuroscience, cognitive psychology, neurotherapeutics, cognitive enhancement, and more. The paper is written clearly enough for the broad audience, is concise and punchy, and has figures that deliver the essential content.

Given my enthusiasm, I only have a few questions that I would like to see the paper answer in a simple revision.

Spectral power is an interesting feature to use to train the classifier, but was anything else tried, such as raw voltage, phase angle, etc.? If there were other features tried that didn't work, this would be useful to slip into the method section to help the next group of pioneers, and those using other techniques (EEG) that would like to follow up. This would increase future citations even more.

What was the timing of the stimulation relative to presentation of the memoranda? Was it fixed so that it was during the interstimulus interval, or was it entirely determined by brain state so that it varied? I could not find this or missed it. The temporal relationship of this would be useful to report (e.g., the mean time of stimulation was X.X ms before the presentation of the next stimulus, +/- X.Xms).

The paper would benefit from including a serial position analysis of the subsequent memory effect relative to when stimulation was delivered. Similar to the temporal contiguity effect, do you see memory for the stimuli after (and perhaps before) the stimulation drop off gradually, or does this just rescue a single stimulus? This would tell people who study

memory whether the stimulation was simply modulating the encoding of new inputs (only effects following stimulation), or changing information storage.

What was the frequency of stimulation with in a stim-list? I would assume there was considerable variability there, but a mean and variance would be useful so that the reader can determine whether it was being tripped every other word, or just a couple of times. This is relevant for the viability of the analysis in the paragraph above.

The paper should just type 'lateral-temporal cortex' instead of using LTC, given that this acronym is not super common in the literature. Best to not have the broad readership of this journal need a decoder ring.

Signed, Geoff Woodman

Responses to Reviewer Comments

We address each reviewer's comments below. Updates to the text are rendered in red in this letter; in the main manuscript we have also highlighted changes in red.

Reviewer #1:

1. *The authors should perform an ANOVA (after checking that the assumptions are met). Or, a non-parametric version of an ANOVA. Factors to include would be stimulation vs. non-stimulation, region, and whether free recall involved categorized or uncategorized words. Given the sample size, it seems that the authors could perform a logistic regression and include single trials as well as a mixed-effects ANOVA with the dependent variable being performance for each condition for each subject. Currently, though the statistics are not clear. The authors should also include information on the effects size.*

We thank the reviewer for raising this issue. One of the suggestions made by the reviewer, performing a logistic regression, is the analysis that we performed within the context of the linear mixed effects framework. Given that this was not obvious to the reader we have revised the following paragraph in the main text:

“We used a generalized linear mixed effects model (logistic regression with a binomial distribution) to model the individual trial-level recalled/not recalled data across subjects to determine how stimulation affected the probability of word recall. The linear mixed effects approach allows us to estimate the effect of stimulation on probability of recall while accounting for heterogeneity in the amount of data collected from each subject. The resulting parameter estimates reflect the change in odds associated with each condition (Stim/NoStim and lateral temporal cortex/Control). Stimulation of lateral temporal cortex increased the odds of recalling stimulated words compared to matched non-stimulated words (odds ratio = 1.18, $P < 0.05$, Fig. 3a) and compared to Control stimulation ($P < 0.01$). Stimulation led to a non-significant decrease in recall odds in the Control group (odds ratio = 0.87 $P = 0.23$). The stimulation targets that led to increased memory performance in the lateral temporal cortex group clustered in the middle portion of the left middle temporal gyrus (Fig. 4). Using a log-binomial model (Barros & Hirakata, 2003) to estimate the relative change in recall probability, we found that lateral temporal cortex stimulation increased the relative probability of item recall by 15% ($P < 0.03$).”

We believe this revision more precisely conveys the statistical approach and findings. With regard to the comment concerning effect size, we believe that this is most accurately achieved by reporting both the odds ratio (which is a measure of effect size) from the logistic regression model and the relative risk from the log-binomial model. By including the additional information on the relative risk, readers will be well-equipped to interpret the effect size in our data.

We also considered performing an ANOVA to further assess the behavioral effects of stimulation, as suggested by the referee, but for several reasons concluded that our dataset was better analyzed using a linear mixed-effects regression. First, ANOVA is not well suited for situations in which there is a mixture of independent and correlated observations. In our data, a subset of subjects were stimulated at more than one target (in separate sessions) and including these observations as

independent datapoints would violate one of the ANOVA assumptions. Second, ANOVA makes the assumption that each observation is equally reliable, i.e. deriving a single measure for each subject for use in ANOVA would mean there would be no information about variance within subject. This assumption is unlikely to be true in our data because some subjects performed a single closed-loop stimulation session while others performed many (up to five for one of our subjects). These features of our dataset are a result of practical issues in intracranial EEG data collection, which occurs in the context of a clinical environment. Linear mixed effects models allow us to explicitly model the correlation structure of the data to account for this heterogeneity, maximize statistical power, and are increasingly recognized as an important tool in the analysis of data collected from intracranial EEG patients (Kadipasaoglu et al., 2014).

2. *Can the authors describe the nature of the stimulation in more detail? The details are relatively sparse but it appears to be gamma-band stimulation which varied in frequency by patient. How was the frequency selected? What were the results on the underlying electrophysiological signals from stimulation? Suppl. Figure 1 would be enhanced by comparing stimulated and non-stimulated trials. The tilt effect is not clear and thus it is not obvious why this is included.*

We agree that the stimulation selection procedure should be described in more detail. The stimulation frequency and amplitude were chosen based on a pre-test conducted within each subject. We have updated the *Stimulation methods* section as follows:

“Stimulation was delivered using charge-balanced biphasic rectangular pulses (pulse width = 300 μ s) at [10, 25, 50, 100 or 200] Hz frequency and [0.25 to 2.00] mA amplitude (0.25 mA steps). The particular amplitude and frequency were chosen based on a pre-test in which we stimulated the brain at each parameter combination while the patient was at rest (no experimental task). The frequency \times amplitude combination that maximized the change in classifier output was used in the closed-loop memory task. In the memory task stimulation was always applied for 500 ms in response to classifier-detected poor memory states (see below).”

We agree with the reviewer that more detail on the electrophysiological effects of stimulation is important for the manuscript. We have therefore conducted an additional analysis to show how the effects of stimulation influence neural activity in regions in which we had good electrode coverage. First, we compared evoked power following stimulation to matched intervals in NoStim lists. To do this we extracted spectral power in the high-frequency range (70–200 Hz) from each electrode following each stimulation event (1000 ms epochs) and for matched periods in unstimulated lists. We compared evoked power in electrodes in the same lobe as the stimulation target between the Stim and NoStim conditions. In Figure 1 we show the average difference in power between the post-stimulation period and matched unstimulated periods for LTC and Non-LTC targets. There was no difference in power for both the LTC and Non-LTC groups, and no difference between them.

This analysis will be of interest to readers because it suggests that stimulation’s effects of neural activity are not homogeneous enough to emerge in an analysis in which *spectral power* is directly aggregated over regions of interest and subjects. It is for this reason that we include the subsequent analysis of the change in classifier output. Each subject’s classifier is an individualized model of successful encoding activity, and it should therefore be more sensitive to changes in neural activity

evoked by stimulation that may be heterogeneous across subjects.

Figure 1: Evoked high-frequency power following stimulation. Within electrodes that were placed in the same lobe and hemisphere as the stimulation target, we extracted wavelet power for 1000 ms windows following stimulation and matched NoStim epochs. We averaged power over time and computed unpaired t-tests within each electrode and wavelet frequency in the high-frequency range (70–200 Hz). We then averaged these t-statistics across electrodes and across subjects to derive an aggregated estimate of the effect of stimulation on high-frequency power near the stimulation target site. Across subjects, high-frequency power was not reliably modulated by stimulation in the same lobe as the stimulation target ($P_{Lat\ temp} = 0.40$; $P_{Control} = 0.99$; $P_{Lat\ temp-Control} = 0.61$). Data are presented as mean \pm s.e.m.

We also updated the caption for Supplementary Figure 1 to highlight the tilt effect in panel (a), namely the simultaneous positive subsequent memory effect (SME) at higher frequencies (> 40 Hz) and negative subsequent memory effect at lower frequencies (3 - 8 Hz). We also now include an analysis of feature importance to in Figure 2 that demonstrates the spectral tilt pattern used by the classifier to differentiate words likely to be recalled and forgotten.

3. The authors mention they also stimulate outside of the lateral temporal cortex. How exactly were these sites selected in terms of subsequent memory effects (largest? What frequency? Etc). Were the site selections always functionally driven? If so, how?

We have updated the *Stimulation methods* section to provide more information about target selection:

“Because the electrode locations were determined strictly by the monitoring needs of the clinicians, we used a combination of anatomical and functional information to select stimulation sites. If available, we prioritized electrodes in lateral temporal cortex, in particular the middle portion of the middle temporal gyrus. To choose among these regions in cases in which more than one was available, we selected the electrode pair demonstrating the largest subsequent memory effect (SME), in the high frequency range (70–200 Hz). In cases in which no lateral temporal cortex contacts were available, we

selected an electrode pair at or near the largest SME elsewhere in the brain, targeting the hippocampus, medial temporal lobe cortex, prefrontal cortex and parietal cortex if available.”

4. The authors argue that their stimulation effects are also whole brain. This is not clear at all from the data presented. What did recording electrodes look like outside of the stimulated area? If the authors wish to argue for whole brain effects, they should probably frame this issue in more detail, which has been discussed in past papers in some detail (e.g., Fell et al. 2013 *Brain Stimulation*, Kim et al. 2016 *NLM*).

The argument we want to make is that the whole-brain classifier is a useful tool for integrating heterogeneous effects of stimulation into a single estimate of the change in memory encoding state. The classifier may be more sensitive to small and distributed changes in activity evoked by stimulation than would, for example, a univariate analysis of spectral power in a local region of interest. To make this more clear, we have revised the following section of the results:

We next asked how lateral temporal cortex stimulation influenced neural activity to support memory encoding by comparing high-frequency activity (HFA, 70-200 Hz) between Stim and NoStim conditions. High-frequency power correlates with single and multi-unit firing (Manning, Jacobs, Fried, & Kahana, 2009), predicts memory success (Long, Burke, & Kahana, 2014; Ezzyat et al., 2017; Burke et al., 2014), and is thought to reflect excitation of neural populations engaged during cognition (Burke, Ramayya, & Kahana, 2015; Miller et al., 2014). When comparing evoked HFA following stimulation to matched unstimulated periods in electrodes placed in the same lobe as the stimulation target, we did not observe significant increases in HFA power following lateral temporal cortex stimulation (Supplementary Fig 2). This suggests that lateral temporal cortex stimulation did not reliably modulate activity in these target regions when aggregating data across subjects, which was not surprising given that stimulation has heterogeneous effects on downstream targets (Haglund, Ojemann, & Blasdel, 1993; Suh, Bahar, Mehta, & Schwartz, 2006) and that our subjects each possessed a unique recording montage. We reasoned that a more sensitive approach would be to assess the effect of stimulation on neural activity using a model that accounts for this between-subject variability.

The classifier that we trained on each subject’s record-only data is an individualized model relating neural activity in that person to a probability of recall, so we next asked whether these classifiers could be used to decode the effects of stimulation on neural activity. We assessed this by computing the *change* in whole-brain classifier output for consecutive words (w_{i+1} and w_i , for all stimulated words i). We did the same for matched words on NoStim lists and compared these Δ classifier values between the *Stim* and *NoStim* conditions for the lateral temporal cortex group using a mixed model. This analysis showed larger increases in classifier output following stimulation compared to the control non-stimulated condition ($estimate = 0.009, P < 0.04$, Fig. 3b), suggesting that stimulation’s influence on neural activity was evident when using the whole-brain classifier to decode the change in neural activity post-stimulation.

MINOR

What was the electrode diameter?

We used several types of electrodes according to the needs of the clinicians at each data collection site, so we have updated the *Methods* to indicate which were used:

“The types of electrodes used for recording and stimulation varied across the data collection sites in accordance with the preferences of the clinicians at each institution. Across the sites, the following models of depth and surface (strips/grid) electrodes were used (electrode diameters in parentheses): PMT Depthalon (0.8 mm); AdTech Spencer RD (0.86 mm); AdTech Spencer SD (1.12 mm); AdTech Behnke-Fried (1.28 mm); AdTech subdural strips and grids (2.3 mm).”

Reviewer #2:

This paper describes findings from an experiment that used intracranial stimulation of patients performing free recall tasks. The goal was to determine whether the lateral temporal cortex is a viable target for stimulation to improve human memory. The experiment worked in a closed-loop manner. Several sessions of intracranial EEG (iEEG) were collected during free recall performance to train a classifier to relate brain state measured with the iEEG activity to subsequent-memory performance. This allowed the algorithm to identify when the brain was in a poor encoding state and intervene with stimulation of the lateral temporal cortex, or stimulation of control sites outside the temporal cortex.

This paper represents an excellent combination of rigorous methods, a unique patient population, and strong behavioral task to measure memory storage. This paper is a bit more exploratory than one in which competing hypotheses are distinguished. But I would expect it to have a large impact (i.e., highly cited) given the propensity for the field to focus on MTL (and the parietal and prefrontal hubs to which it connects). This paper is highly suitable for Nature Communications as it represents a new discovery with impacts in the fields of neurology, neuroscience, cognitive psychology, neurotherapeutics, cognitive enhancement, and more. The paper is written clearly enough for the broad audience, is concise and punchy, and has figures that deliver the essential content.

Given my enthusiasm, I only have a few questions that I would like to see the paper answer in a simple revision.

We thank the reviewer for their enthusiasm.

Spectral power is an interesting feature to use to train the classifier, but was anything else tried, such as raw voltage, phase angle, etc.? If there were other features tried that didn't work, this would be useful to slip into the method section to help the next group of pioneers, and those using other techniques (EEG) that would like to follow up. This would increase future citations even more.

While we did not mention this in the original manuscript, we informally evaluated using phase-based measures of connectivity as features for classification. We have updated the *Multivariate classification* section of the methods to reflect this:

We used spectral power averaged across the time dimension for each word encoding epoch (0–1366 ms relative to word onset) as the input data (informal investigations suggested that memory classification with spectral power outperformed features derived from phase-based connectivity measures).

What was the timing of the stimulation relative to presentation of the memoranda? Was it fixed so that it was during the inter-stimulus interval, or was it entirely determined by brain state so that it varied? I could not find this or missed it. The temporal relationship of this would be useful to report (e.g., the mean time of stimulation was X.X ms before the presentation of the next stimulus, +/- X.Xms).

We have updated the *Classification methods* to clarify the timing of stimulation:

Closed-loop decoding

The closed-loop free recall sessions consisted of one practice list followed by 25 task lists: lists 1-3 were unstimulated and used (in addition to the practice list) to collect baseline spectral power data for normalizing the input features to the classifier; lists 4-25 consisted of 11 lists each of Stim and NoStim conditions, randomly interleaved. On Stim lists, we extracted estimates of spectral power from the 0–1366 ms for each word encoding period. We assessed power using parameters identical to those used to train the record-only classifier (see *Electrophysiological data processing*): 8 frequencies from 3–180 Hz, logarithmically-spaced; Morlet wavelets; wave number = 5; log-transformed; averaged over time within frequency and electrode; and z-scored based on the mean and standard deviation of the power features collected during the baseline lists. The distribution of powers used for normalization was updated following each NoStim list. The record-only classifier was applied to the resulting frequency × electrode features to derive an estimated probability of recall. If this probability fell below 0.5, we immediately triggered 500 ms of stimulation (mean interval between stimulation onset and presentation of the next word = 962 ms ± 347 ms). On NoStim lists, spectral power features and classifier output were computed identically to Stim lists, but stimulation was disabled.

The paper would benefit from including a serial position analysis of the subsequent memory effect relative to when stimulation was delivered. Similar to the temporal contiguity effect, do you see memory for the stimuli after (and perhaps before) the stimulation drop off gradually, or does this just rescue a single stimulus? This would tell people who study memory whether the stimulation was simply modulating the encoding of new inputs (only effects following stimulation), or changing information storage.

We conducted an analysis comparing the proportion of words recalled for Stim and NoStim lists between the LTC and Non-LTC groups. For each Stim word, we identify whether it was preceded and followed by an unstimulated word (we do the same for NoStim lists by matching trials based on classifier output). We then compare recall for these flanking items separately for cases in which

the center item was not recalled. This is important to control for the fact that Lateral temporal cortex stimulation increased recall for stimulated items compared to Control stimulation, which would be expected to lead to differences in memory for neighboring items even if stimulation only affected the stimulated item (Kahana, 2012). We first modeled main effects and interactions of Stim/NoStim, Lag (± 1) relative to the stimulated word, and Lateral Temporal/Non-lateral temporal group. There were no main effects or interactions with Lag (all $P > 0.74$), so we collapsed the forward and reverse directions in a linear mixed model comparing recall on neighboring items as a function of Stim/NoStim and Lateral Temporal/Non-lateral temporal group. Stimulation differentially affected memory for neighboring items (Stim/NoStim \times Lateral temporal/Non-lateral temporal interaction $P < 0.02$). This effect was driven by a significant increase in memory for the Lateral temporal stimulation group ($P < 0.02$).

We now describe this analysis in the manuscript results section.

What was the frequency of stimulation with in a stim-list? I would assume there was considerable variability there, but a mean and variance would be useful so that the reader can determine whether it was being tripped every other word, or just a couple of times. This is relevant for the viability of the analysis in the paragraph above.

To address this question we assessed (1) the overall proportion of words stimulated for the LTC and Non-LTC groups; and (2) the serial position curve for the proportion of words stimulated. This analysis showed that stimulation was applied on roughly half of trials for both groups (LTC = 0.53 ± 0.02 ; Non-LTC = 0.52 ± 0.04 ; $t(27) = 0.38, P = 0.71$), and did not differ as a function of serial position between the two groups.

The paper should just type "lateral-temporal cortex" instead of using LTC, given that this acronym is not super common in the literature. Best to not have the broad readership of this journal need a decoder ring.

We have taken this suggestion and have replaced LTC with *Lateral temporal cortex* and Non-LTC with *Non-lateral temporal* throughout the manuscript.

References

- Barros, A. J., & Hirakata, V. N. (2003). Alternatives for logistic regression in cross-sectional studies: an empirical comparison of models that directly estimate the prevalence ratio. *BMC medical research methodology*, 3, 21.
- Burke, J. F., Long, N. M., Zaghoul, K. A., Sharan, A. D., Sperling, M. R., & Kahana, M. J. (2014). Human intracranial high-frequency activity maps episodic memory formation in space and time. *NeuroImage*, 85, 834–843.
- Burke, J. F., Ramayya, A. G., & Kahana, M. J. (2015). Human intracranial high-frequency activity during memory processing: Neural oscillations or stochastic volatility? *Current Opinion in Neurobiology*, 31, 104–110.
- Ezzyat, Y., Kragel, J. E., Burke, J. F., Levy, D. F., Lyalenko, A., Wanda, P., . . . Kahana, M. J. (2017). Direct brain stimulation modulates encoding states and memory performance in humans. *Current Biology*, 27, 1-8.
- Haglund, M. M., Ojemann, G. A., & Blasdel, G. G. (1993). Optical imaging of bipolar cortical stimulation. *Journal of Neurosurgery*, 78, 785–793.
- Kadipasaoglu, C., Baboyan, V., Conner, C. R., Chen, G., Saad, Z. S., & Tandon, N. (2014). Surface-based mixed effects multilevel analysis of grouped human electrocorticography. *Neuroimage*, 101, 215–224.
- Kahana, M. J. (2012). *Foundations of human memory*. New York, NY: Oxford University Press.
- Long, N. M., Burke, J. F., & Kahana, M. J. (2014). Subsequent memory effect in intracranial and scalp EEG. *NeuroImage*, 84, 488–494. doi: 10.1016/j.neuroimage.2013.08.052
- Manning, J. R., Jacobs, J., Fried, I., & Kahana, M. J. (2009). Broadband shifts in local field potential power spectra are correlated with single-neuron spiking in humans. *Journal of Neuroscience*, 29, 13613–13620.
- Miller, K. J., Honey, C. J., Hermes, D., Rao, R. P., den Nijs, M., & Ojemann, J. G. (2014, January). Broadband changes in the cortical surface potential track activation of functionally diverse neuronal populations. *NeuroImage*, 85, 711–720.
- Suh, M., Bahar, S., Mehta, A. D., & Schwartz, T. H. (2006). Blood volume and hemoglobin oxygenation response following electrical stimulation of human cortex. *Neuroimage*, 31, 66–75.

REVIEWERS' COMMENTS:

Reviewer #1 (Remarks to the Author):

The authors have addressed most of my concerns, although they have introduced some small issues that should be corrected during the revision.

1) In some places, the authors just report p values. Should these have some accompanying statistical values with them, like F or beta values?

2) "Stimulation also did not differ between the groups as a function..."

Incomplete sentence here

3) "High-frequency power correlates with single and multi-unit firing [36]"

There is a much richer literature on this topic than cited here and this correlation will not be true for the larger diameter electrodes used in this study (the Manning et al. study cited here used microelectrodes). In fact, these past high-frequency power multi and single unit firing rate correlations vary by region and are often not observed in the medial temporal lobes. This statement, as given in the current version of the manuscript, is thus incorrect and should be amended to reflect the multifaceted nature of the LFP.

4) "This analysis showed increases in classifier output following stimulation"

did the authors remove stimulation artifact here? Otherwise, the classification would appear trivial.

Reviewer #2 (Remarks to the Author):

The authors have done an excellent job of addressing the comments from my initial review. I have no concerns about the suitability of the paper for publication. This is an exciting set of results and will be of interest to a broad readership.

Signed, Geoff Woodman

Response to Reviewers

Reviewer 1:

The authors have addressed most of my concerns, although they have introduced some small issues that should be corrected during the revision.

1) In some places, the authors just report p values. Should these have some accompanying statistical values with them, like F or beta values?

We have addressed this point, now including the statistical tests in each case.

2) “Stimulation also did not differ between the groups as a function...” Incomplete sentence here

We have confirmed that this sentence is now complete.

3) “High-frequency power correlates with single and multi-unit firing [36]”

There is a much richer literature on this topic than cited here and this correlation will not be true for the larger diameter electrodes used in this study (the Manning et al. study cited here used microelectrodes). In fact, these past high-frequency power multi and single unit firing rate correlations vary by region and are often not observed in the medial temporal lobes. This statement, as given in the current version of the manuscript, is thus incorrect and should be amended to reflect the multifaceted nature of the LFP.

We have amended the sentence to read as follows: “High-frequency power predicts memory success^{1,37,40,41} and is thought to reflect excitation of neural populations engaged during cognition^{42,43}.”

**4) “This analysis showed increases in classifier output following stimulation”
did the authors remove stimulation artifact here? Otherwise, the classification would appear trivial.**

For this analysis the classifier was applied to data after stimulation ended (with an intervening buffer), thus avoiding artifacts from the stimulation itself. The increased classifier output for these intervals therefore reflects increased evidence for good encoding states following stimulation, not a direct comparison between stimulated and unstimulated epochs.

Reviewer 2:

The authors have done an excellent job of addressing the comments from my initial review. I have no concerns about the suitability of the paper for publication. This is an exciting set of results and will be of interest to a broad readership.

Signed, Geoff Woodman

We thank the Reviewer for their helpful comments.